# Searching for the Novel Specific Predictors of Prostate Cancer in Urine: The Analysis of 84 miRNA Expression

**DOI:** 10.3390/ijms19124088

**Published:** 2018-12-17

**Authors:** Evgeniy A. Lekchnov, Evgeniya V. Amelina, Olga E. Bryzgunova, Ivan A. Zaporozhchenko, Mariya Yu. Konoshenko, Sergey V. Yarmoschuk, Ivan S. Murashov, Oxana A. Pashkovskaya, Anton M. Gorizkii, Aleksandr A. Zheravin, Pavel P. Laktionov

**Affiliations:** 1Institute of Chemical Biology and Fundamental Medicine, Siberian Branch, Russian Academy of Sciences, Novosibirsk 630090, Russia; olga.bryzgunova@niboch.nsc.ru (O.E.B.); ivanzap@niboch.nsc.ru (I.A.Z.); lacyjewelrymk@gmail.com (M.Y.K.); lakt@niboch.nsc.ru (P.P.L.); 2Meshalkin Siberian Federal Biomedical Research Center, Ministry of Public Health of the Russian Federation, Novosibirsk 630055, Russia; s_jarmoschuk@meshalkin.ru (S.V.Y.); ivmurashov@gmail.com (I.S.M.); oxana.pashkovskaya@gmail.com (O.A.P.); a_goritskij@meshalkin.ru (A.M.G.); a_zheravin@meshalkin.ru (A.A.Z.); 3The Center for Technology Transfer and Commercialization, Novosibirsk State University, Novosibirsk 630090, Russia; amelina.evgenia@gmail.com

**Keywords:** miRNA, urine, extracellular vesicles, prostate cancer

## Abstract

The aim of this study was to investigate miRNA profiles of clarified urine supernatant and combined urine vesicle fractions of healthy donors and patients with benign prostatic hyperplasia and prostate cancer (PCa). The comparative analysis of miRNA expression was conducted with a custom miRCURY LNA miRNA qPCR panel. Significant combinations of miRNA pairs were selected by the RandomForest-based feature selection algorithm Boruta; the difference of the medians between the groups and a 95% confidence interval was built using the bootstrap approach. The Asymptotic Wilcoxon-Mann-Whitney Test was performed for miRNA combinations to compare different groups of donors. Benjamini-Hochberg correction was used to adjust the statistical significance for multiple comparisons. The most diagnostically significant miRNAs pairs were miR-107-miR-26b.5p and miR-375.3p-miR-26b.5p in the urine supernatant fraction that discriminated the group of healthy patients and PCa patients, as well as miR-31.5p-miR-16.5p, miR-31.5p-miR-200b, miR-31.5p-miR-30e.3p and miR-31.5p-miR-660.5p in the fraction extracellular vesicles that were different between healthy men and benign prostate hyperplasia patients. Such statistical criteria as the occurrence of individual significant miRNA pairs in the total number of comparisons, median Δ*C*_t_ difference, and confidence interval can be useful tools for determining reliable markers of PCa.

## 1. Introduction

Prostate cancer (PCa) is rightfully considered a global social problem because of its prevalence [1] and high associated mortality in developing countries [2]. According to the American Cancer Society 2017, in developed countries, the relative 5-year survival rate for localised PCa is nearly 100%, but at advanced stages it is only about 29% [3]. Similar to many other cancers, lack of early symptomatic manifestation is a major contributor to the late detection of PCa. Modern PCa diagnostics are based on the detection of prostate specific antigen (PSA) in blood, digital rectal examination (DRE) and transrectal ultrasonography (TRUS). The final diagnosis is based upon the results of histological analysis of bioptates or postoperative tumour samples. Additional survey methods including computed tomography scans (CT), different variations of magnetic resonance imaging (MRI), positron emission tomography (PET), X-rays and skeletal scintigraphy are used to assess the tumour spread and determine the stage of the disease. However, the sensitivity of the first-line diagnostic procedures—DRE and TRUS is low. For example, DRE is positively correlated with tumour stage in <50% of cases and TRUS cannot differentiate between T2 and T3 stage tumours with sufficient accuracy [4]. Moreover the effectiveness of TRUS interpretation greatly depends on the experience and qualification of the sonographer. The sensitivity of CT and MRI is also insufficient. Thus, MRI is not recommended for local staging in patients with low risk PCa (PSA < 10 ng/mL, and Gleason score < 7 and T1-2a stages) [4]. Besides, the use of CT scans and MRI for PCa screening is considered impractical from an economic standpoint. Currently, PSA is the most widely used screening marker for PCa but it suffers from low specificity. With 70–90% sensitivity, the specificity of a blood PSA test is only 20–40% (Area Under the ROC Curve (AUC) 0.55–0.70) [5]. PSA does not allow us to distinguish an inflammatory process in the prostate from the benign tissue growth or to determine whether PCa is running an aggressive or indolent course [6]. Nevertheless, the specificity of PSA is higher than that of DRE and TRUS [4]. The use of PSA for PCa screening reduced the number of the patients diagnosed with advanced disease [7], but the benefit is counterbalanced by overdiagnosis, detection of latent PCa, clinically insignificant benign tumors and, as a consequence, overtreatment of patients. In this light, the feasibility of PSA as a biomarker of PCa is actively debated [8]. Due to this, PSA does not meet modern criteria for an effective cancer biomarker and a replacement is overdue. It is therefore necessary to develop a more reliable, highly specific and sensitive test, based on the use of novel diagnostic, prognostic and predictive biomarkers to assist with PCa diagnosis and staging, definition of risk groups, evaluation of tumor response to treatment, monitoring of drug resistance, timely detection of relapses and metastases [9].

Recently, the methods of molecular biology have been extensively adopted for use in diagnostics of cancer, including PCa [5,10]. One of the modern approaches is to search for novel cancer biomarkers among circulating cell-free nucleic acids in plasma and urine. Such markers can potentially be tumour-specific mRNA, long non-coding RNA, aberrant DNA methylation profiles, specific mutations and rearrangements, and overexpressed miRNA [10,11]. By now, several test-systems for PCa diagnostics have been proposed. Detection of long non-coding RNA PCA3 in urine pellets was shown to give higher diagnostic performance than PSA assays (0.69 AUC for PCA3 and 0.55 AUC for PSA [12] and can be used to determine whether repeated biopsy after negative primary biopsy is required [4]. However, the best discriminating value for PCA3 has not yet been determined, which may influence the percentage of PCa cases diagnosed using this test [13]. Another PCa marker is the fusion transcript TMPRSS2:ERG which is overexpressed in epithelial PCa cells. When measured in urine this marker offers a high specificity (93%), but the sensitivity is low (37%) [6]. Moreover, so far no correlations between TMPRSS2:ERG expression and either Gleason score or clinical stage of PCa were detected, thus making it a less desirable prognostic tool [13].

Diagnostic systems using a combination of different markers can improve analytical characteristics of PCa detection. Several such diagnostics incorporating either PSA or PCA3 were recently approved by the FDA (US Food and Drug Administration). Some combination tests include PSA analysis in blood (4K score test, Prostate health index, Mi-Prostate score test) while the others require biopsy or surgical material for analysis (Oncotype DX test, ConfirmMDx test, ProMark test) [6,13,14]. Still, despite a variety of available tests, options for effective diagnosis and prognosis of PCa are still lacking [15].

There are several obvious benefits of using biological fluids as a source of biomarkers. First is the ability to capture the genetic heterogeneity of the entire tumor, in contrast to tumor tissue fragments from specific areas obtained by biopsy [1,10]. The drawbacks are that liquid biopsy often fails to differentiate between primary and metastatic tumor sites and that due to the low concentration of diagnostic molecules, a highly sensitive detection is required [10]. The search for liquid biopsy biomarkers which will allow us to detect PCa with a high degree of reliability still continues.

Extracellular miRNAs found in biological fluids are considered as one class of perspective PCa markers. It was shown that these molecules take part in the key processes of cancerogenesis, and their expression is determined by the tumor status, i.e., growth rate, the tendency of the tumor to metastasize, etc., [1,9,11,15,16,17,18]. Moreover, miRNAs can stably persist in biological fluids for extended periods of time [19], which ensures their reliable detection. The analysis of miRNA expression requires a small amount of biological material (obtained in less-invasive or non-invasive manner) and can be performed with a minimum set of equipment and few personnel by using such well-developed methods as microarray analysis, sequencing and RT-PCR. These properties instill hope that a set of diagnostically significant miRNAs can be discovered in blood or urine that can be used to develop a panel of markers for a more sensitive and specific PCa diagnosis, suitable for implementation in routine clinical practice.

In this study, we investigated the miRNA profiles of clarified urine supernatant and combined urine extracellular vesicles (EVs) in groups of healthy men and patients with benign prostatic hyperplasia (BPH) and PCa to identify miRNAs that can allow us to reliably distinguish PCa patients from non-cancer groups, and evaluate the diagnostic potential of selected predictors.

## 2. Results

The threshold cycle (*C*_t_) values obtained after miRNA profiling were normalized using the pair ratio method, effectively evaluating the expression of all possible combinations of two miRNAs [20,21]. For the analyses, the following comparison groups were established: healthy donors–prostate cancer patients (comparison groups h–c for urine supernatant and hm–cm for urine EVs), healthy donors–patients with BPH (comparisons h–b and hm–bm) and patients with BPH–PCa patients (comparisons b–c and bm–cm).

Next, data on the occurrence of individual significant miRNA pairs in the total number of comparisons (frequency), median ∆*C*_t_ difference (median distance), confidence interval (CI) and unadjusted (*p*) and adjusted (*p*_adj_) significance were collected using the Boruta feature selection method as described in “Materials and Methods”. The best miRNA combinations (Table 1) were selected based on the following criteria: 3 pairs with the highest frequency; 3 pairs with the largest median distance between the groups of donors (but no less than 1 *C*_t_); 3 pairs with 95% CI of the median distance most removed from 0 (but no less than 0.1); 3 pairs with the lowest value of *p*_adj_. Different criteria allow for flexibility in approaching the group comparisons. Supposedly, the frequency criterion is a valuable indicator provided by the Boruta algorithm that allows for selecting of miRNA pairs that are less sensitive to sampling bias and can be effectively combined with other pairs. The median distance and the 95% CI criteria reflect the “diversity” of values in different groups and are a measure of practical applicability. If group value clusters are in close proximity to each other, small errors in the miRNA expression measurement (e.g., technical and sampling variation) can dramatically affect the accuracy of group identification and the quality of the analytical system. Pairs with the lowest value of *p*_adj_ were selected as the most differently expressed from the statistical point of view. In general, since these criteria aimed at achieving the most accurate separation of groups, the pairs identified by the criteria often overlapped. In this case, any pair suggested by several criteria was only included in the final selection once. For each of the selected miRNA pairs, ROC analysis was performed and AUC was determined.

It is noteworthy that the frequency of occurrence in Boruta runs was markedly different between the miRNA pairs: miR-107-miR-26b-5p (74.8%, h–c), hsa.miR.30a.5p-hsa.let.7g.5p (98.5%, h–b), miR-20a.5p-miR-16.5p (50.3%, hm–cm) and miR-107-miR-31.5p (95.3%, hm–bm). The AUC values for the presented pairs of miRNAs ranged from 0.70 to 0.93.

In urine supernatant for the h–c group comparison, the best pairs were: hmiR-375-miR-26b.5p, miR-29a.3p-miR-205.5p, miR-331.3p-miR-205.5p, miR-205.5p-miR-26b.5p, miR-29b.3p-miR-205.5p and miR-151a.5p-miR-205.5p. The miR-29b.3p-miR-205.5p pair had the greatest median distance while for miR-29a.3p-miR-205.5p, the CI of the median difference was most removed from zero. The miR-107-miR-26b.5p was the most frequently occurring pair (74.8%), but it had a Δ*C*_t_ difference of less than 1. The h–b and b–с comparisons only yielded 3 and 6 miRNA pairs, respectively, with borderline significant differences. This is partially because several miRNAs expressed close to the assay sensitivity level were not be detected in some of the samples from these groups. When comparing healthy donors and BPH patients, the only significant pair was miR-30a.5p-let.7g.5p with a high frequency level of 98.5%. The pair miR-103a.3p-miR.30c.5p had the highest occurrence (28.6%) and median difference (1.03) in the b–с comparison.

In the comparison of miRNA pairs in the EV fraction of healthy donors and PCa patients (hm–cm) the most frequent pair was miR-20a.5p-miR-16.5p (50.3%), but its median distance was only 0.39. The greatest median difference was obtained for miR-31.5p-miR-16.5p (0.77) and miR-30b.5p-miR-16.5p (0.85). The miRNA pair miR-29a.3p-miR-30e.3p was characterized by the greatest median difference (0.82) in the hm–bm comparison (EV fraction), but its CI was too close to 0. The most frequent miRNA pair in this comparison was miR-107-miR-31.5p (95.3%). In the bm–cm comparison, the miR-191.5p-miR-31.5p combination had the highest occurrence, while miR-106b.5p-miR-191 was characterized by the greatest Δ*C*_t_ difference (0.71)**.**

It is worth noting that, despite the uniqueness of predictor pairs for each of the 6 comparisons, the same miRNAs were detected simultaneously in pairs from several comparisons (Table 2). Interestingly, some miRNAs were detected only in the urine supernatant (miR-23b.3p, miR-30a.5p, miR-205.5p), while others exclusively in the EV fraction (miR-24.3p, miR.31.5p and miR-200b.3p).

In each comparison, certain miRNAs repeatedly occurred in different combinations (Table 3). For example, miR-205.5p and miR-26b.5p were present in 50% and 40% of pairs, respectively in the h–c comparison. The miR-30c.5p was in each of the three miRNA pairs in the h–b comparison, and miR-31.5p was seen in 6 of the 10 combinations in hm–bm.

The sensitivity and specificity of group separation were determined for the selected pairs, revealing the diagnostic efficacy of the following miRNA pairs: miR-107-miR-26b.5p (AUC 0.93, 80% sensitivity and 100% specificity), miR-375.3p-miR-26b.5p (AUC 0.83, 70% sensitivity and 100% specificity) for h–c; miR-31.5p-miR-16.5p (AUC 0.89, 80% sensitivity and 100% specificity), miR-31.5p-miR-200b (AUC 0.88, 70% sensitivity and 100% specificity), miR-31.5p-miR-30e.3p (AUC 0.88, 80% sensitivity and 100% specificity) and miR-31.5p-miR-660.5p (AUC 0.84, 70% sensitivity and 100% specificity) for hm–bm, Table 4.

The distribution diagrams of normalized expression values of miRNA pairs were plotted to estimate the group separation of the most promising miRNA pairs (Figure 1). As shown in Figure 1A, the values of the miR-107-miR-26b.5p pair in PCa patients and healthy donors were tightly spread and partially overlap, which can negatively affect group separation, and even more so in the case of measurement errors. By comparison, the likelihood of discriminating healthy donors from PCa patients using miR-375.3p-miR-26b.5p expression is significantly higher (Figure 1A).

A similar pattern could be seen in the hm–bm comparison, where the values of miR-31.5p-miR-660.5p and miR-31.5p-miR.30e.3p pairs were sufficiently well spread, while miR-31.5p-miR-16.5p and miR-31.5p-miR-200с values of BPH patients overlapped with the values for healthy donors, which prevents the efficient separation of these groups (Figure 1B).

Thus, here we compared miRNA expression in groups of healthy donors and patients with BPH and PCa using various statistical criteria to identify the best predictors that could reliably distinguish all three groups. In general, the largest number of such miRNA pairs (*p*_adj_ < 0.05) was found in the comparison group of healthy donor-PCa patients in the urine supernatant fraction (h–c, 6 miRNA pairs) and in the urine EV fraction (hm–cm, 3 miRNA pairs). Additionally, the urine supernatant fraction had more pairs that reliably distinguish healthy donors and PCa patients with Δ*C*_t_ from 1 to 2 (Table 1) than the EV fraction. The most diagnostically significant pairs for discriminating between healthy donors vs. PCa patients were miR-107-miR-26b-5p and miR-375-3p-miR-26b-5p in the urine supernatant fraction, (Table 4). When comparing the groups of healthy donors vs. BPH patients or BPH patients vs. PCa patients, it was not possible to find miRNA pairs that could effectively distinguish donors from the two groups due to borderline statistical significance (urine supernatant fraction) or weak statistical criteria (EV fraction), Table 1. However, when assessing the diagnostic value of predictors in the group of healthy donors vs. BPH patients, the largest number of diagnostically significant pairs was found in the urine EV fraction (miR-31-5p-miR-16-5p, miR-31-5p-miR-200b, miR-31-5p-miR-30e-3p and miR-31-5p-miR-660-5p), Table 4.

## 3. Discussion

Currently, there exists a range of unresolved research challenges concerning miRNA investigation in biological fluids, including urine. For example, there are no universal well-defined standards for preanalytical, analytical and postanalytical stages of biomarker investigation, including protocols of the collection, processing and storage of biological samples, miRNA isolation procedures, biomarker quantification routines, data analysis, etc., [9]. Such a lack of clarity leads to a situation where results from different studies are difficult to compare, and reduces the value of meta-analyses [9]. Despite these existing problems, urine is still considered a promising source of PCa biomarkers, due to the non-invasiveness of sample collection and confirmed presence of PCa specific molecules [12]. 

The differences in miRNA content in different urine fractions (Table 2) may be related to unequal distribution of certain miRNAs between cell-free protein and lipoprotein complexes and EVs [22]. Thus, any of these urine fractions can potentially be used to search for diagnostic and prognostic miRNAs [23,24,25,26,27,28,29,30,31,32,33,34,35,36]. In view of contradicting results of previous investigations, the question of which urine fraction is better suited for miRNA biomarker discovery remains open.

EVs are one of the most extensively researched sources of cancer biomarkers, containing markers of different nature, ref. [37] including tumour-specific miRNA [34,35,36]. Urine EVs are a valuable source of miRNA because of the better signal to background ratio than in cells of urine sediments [38] and higher relative miRNA concentration in comparison with cell-free urine [15]. However, EV isolation by standard methods is more time-consuming and laborious than preparation of cell sediments or cell-free urine samples. Moreover, isolation of urine EVs miRNA is complicated by the presence of uromodulin (Tamm-Horsfall protein). Polymerization of uromodulin molecules can create structures that are capable of entrapping the EVs, reducing their isolation efficiency [39] and also contaminating extracted miRNA samples [40]. That is why in the present work both the urine EVs and cell-free urine supernatant were investigated in search for diagnostically valuable miRNAs (Table 1). In previously published work it was suggested that the changes in miRNA signature in different urine fractions can reflect different pathological processes in the prostate [22]. Thus, simultaneous investigation of miRNA expression in several urine fractions could be of considerable scientific and practical interest.

There are various normalization procedures used for miRNA expression analysis [41]. Among them, the ratio normalization method is a simple and robust approach in which all possible combinations of every two miRNAs are constructed and screened for links to pathological processes, in our case, PCa. In this approach, some of the frequently occurring miRNAs (Table 3) can be assumed to be oncospecific, whereas rarely occurring paired with them are potential stable normalizers. As such, this approach allows us to select a complete panel of oncospecific miRNAs.

It is known that miRNAs play important roles in the key aspects of PCa carcinogenesis and development, including overexpression of androgen receptor (AR), apoptosis resistance, loss of cell cycle control, cell adhesion and epithelial-mesenchymal transition [16,17,18,42,43]. These events are some of the major steps in the acquisition of invasive properties by PCa cells, giving them the propensity to proliferate uncontrollably, enhanced cell survival, mobility and the ability to spread to other organs and tissues. Thus, the detection of oncogenic miRNAs, involved in different pathways of PCa carcinogenesis, can be used as a diagnostic and treatment monitoring tool for this disease. For example, previous studies have shown that miR-205 and miR-214 expression in urine correlated with PCa progression [28] and miR-16, miR-21, miR-222 can be used as predictors of aggressive PCa [30].

In order to access the possible involvement of miRNAs in PCa cancerogenesis, miRNAs from pairs listed in Table 1 were analysed using OncomiRDB and DIANA-TarBase, containing experimentally validated oncogenic and oncosupressor miRNAs. The miR-let-7a and miR-let-7g from the h–b comparison were excluded based on low significance values (*p*_adj_ > 0.05). Most of the remaining miRNAs were identified by both databases. According to OncomiRDB and DIANA-TarBase, 52.8% and 65% of these miRNAs were found to be prostate-cancer specific and were previously implicated in PCA development. The results of the miRNA search in OncomiRDB are shown in Table 5. Moreover, the most frequently occurring miRNAs in each group of comparison could also be associated with PCa development (Table 3), which can be taken into account in developing analytical diagnostic systems. 

Interestingly, according to this data, miRNA pairs identified in h–b and hm–bm comparisons contained oncogenic miRNAs (Table 2). Moreover, the differences between the expression of miRNA pairs in urine supernatants of BPH and PCa patients were only borderline significant, suggesting similar miRNA expression profiles (Table 1). This can be due to the mosaic heterogeneity of prostate tumours–non-diagnosed malignant foci can be present in patients with BPH, and areas of benign growth persist in PCa. BPH is also known to atypically progress into intraductal dysplasia (PIN) accompanied by the alteration or loss of characteristic tissue structure. Although no signs of malignant transformation are generally present in PIN, it can be considered as a pre-cancerous state [44]. Moreover, according to recent data, up to 25 percent of BPH cases are later discovered to have PC [5]. This supports older data that latent prostate carcinomas accompany 15.1% of BPH cases [45]. Another study revealed that nearly one third all cancerous lesions in the transitional zone of the prostate co-exist with BPH [46]. Also there are a number of pathophysiological similarities between BPH and PCa [47,48], including age-dependency profile and androgen requirement for growth and development. Still, to date no explicit evidence linking the two pathologies has been presented [44]. In this light, it is possible that in both BPH and PCa, changes in the expression of some miRNA may be tissue-specific, rather than tumor-specific, as is the case with PSA.

Here, after miRNA expression profiling in three donor groups, we were unable to find individual miRNAs pairs that could accurately distinguish between all three groups Nevertheless, 100% sensitivity and specificity can be achieved by using various combinations of miRNAs. For example, separation of healthy donors and PCa patients using miRNA pairs miR-107-miR-26b.5p, miR-375-miR-26b.5p individually achieved 100%/80% and 70%/79% specificity/sensitivity, respectively. When the pairs were used simultaneously, the sensitivity increased to 100% (Table 4). The same holds for separation of healthy donors and BPH patients (Table 4). The combination of miR-31.5p-miR-16.5p and miR-20a.5p-miR-16.5p pairs in hm–bm increased the sensitivity to 100% instead of 80% and 70%, respectively. Because of similarities in miRNA expression of patients with PCa and BPH, data of miRNA expression in both urine fractions were used to gain 100% sensitivity and specificity of group separation. For example (Table 4): miR-103a.3p-miR-30c.5p (b–c, 60% sensitivity), miR-100.5p-miR-200b.3p (bm–cm, 50% sensitivity) and let-7i.5p-let-7a.5p (bm–cm, 40% sensitivity).

One distinctive feature of miRNA expression in biological fluids is that the magnitude differences between cancer patients and control groups tend to be lower or comparable with the within group variation. This fact calls for strict requirements to be placed on the design of a diagnostic system, including the use of spike-in controls, a certain number of analyzed replicates, and straightforward inclusion and exclusion criteria for miRNA samples, isolated from blood and urine. Only after the formulation of such requirements and development of analytical systems that satisfy them, would it be possible to start the verification of miRNA markers and determine the effectiveness of miRNA-based diagnostic systems.

## 4. Materials and Methods

### 4.1. Study Population and Sample Collection

Blood (used only to determine the PSA level) and urine samples of 10 healthy male individuals (HD), 10 patients with BPH and 10 previously untreated PCa patients were obtained from the Center of New Medical Technologies of ICBFM SB RAS and Regional Oncology Center (Novosibirsk, Russia).

Clinicopathological and demographic characteristics of donors are presented in Table 6. Biological samples were harvested from 10 healthy donors (HD), 10 patients with benign prostatic hyperplasia (BPH) and 10 prostate cancer patients (PCa with T_2-3_NxMx stage and pathological Gleason score 6–7). None of the patients had undergone surgical treatment or received chemotherapy prior to/at the time of sample collection.

The work was conducted in compliance with the principles of voluntariness and confidentiality in accordance with the “Fundamentals of Legislation on Health Care”. The study was approved by the ethics committees of ICBFM SB RAS and Novosibirsk Regional Oncology Center, (minutes of meeting N 10 from 22 December 2008) and written informed consent was provided by all participants.

### 4.2. Urine Fractionation and Isolation of Extracellular Vesicles

To pellet cells, fresh urine was clarified by centrifugation at 400× *g*, 20 °C, for 20 min within 3 h after collection. Supernatants were then centrifuged at 17,000× *g*, 20 °C, for 20 min. Aliquots from supernatants were immediately frozen and stored at −20 °C and thawed once immediately before use.

Total extracellular vesicles (EVs) were precipitated from the 17,000× *g* supernatant by high-speed centrifugation at 100,000× *g*, 18 °C, for 90 min. The pellets were washed with 10 mL PBS, resuspended in 100–300 μL of PBS and frozen in liquid nitrogen and stored at −80 °C until use. Aliquots were thawed once immediately before use.

### 4.3. miRNA Isolation and Analysis

For miRNA extraction from EVs, a sample of 200 μL resuspended EVs was sequentially mixed with 100 μL of denaturing buffer (1% 2-mercaptoethanol and 0.3 М guanidine isothiocyanate and 20 mM Tris–acetate, pH 4) and 200 µL of precipitation buffer (12 mM OcA and 0.8 M sodium acetate pH 4.0). The mixture was vortexed for 5 s, incubated for 5 min at room temperature and centrifuged at 16,200× *g*. Supernatant was mixed thoroughly with an equal volume of 95% ethanol and applied to a silica spin column (e.g., BioSilica Ltd., Novosibirsk, Russia). Then, miRNA extraction was performed as described previously [49]. The modified acid phenol–chloroform extraction was used for miRNA isolation from urine supernatant samples [50]. A sample of 2 mL urine was sequentially mixed with 2 mL of denaturing buffer (1% 2-mercaptoethanol and 3 М guanidine isothiocyanate) and 75 μL of 2 M sodium acetate, pH 4. Then, 4 mL of phenol and 700 μL chloroform were sequentially added and mixed thoroughly. The mixture was centrifuged at 9000× *g*, 4 °С for 20 min. After phase separation, the water phase was collected and phenol extraction was re-performed. The water phase obtained after the second extraction was mixed with an equal volume of 2 M sodium acetate, pH 4, double volume of 96% ethanol, and purified using silica columns (BioSilica Ltd., Novosibirsk, Russia). The column was washed with washing buffer 1 (0.3 M guanidine isothiocyanate, 10 mM Tris–acetate, pH 6.5, 50% ethanol, 1% 2-mercaptoethanol) and washing buffer 2 (10 mM Tris–HCl, pH 7.5, 0.1 M NaCl, 75% ethanol). RNA was eluted from the column with BioSilica RNA elution solution (BioSilica Ltd., Novosibirsk, Russia).

### 4.4. Analysis of miRNA Expression

The comparative analysis of miRNA expression was conducted with a custom miRCURY LNA miRNA qPCR panel (Exiqon, Danmark) based on 67 miRNA samples from a pre-formed Urine Exosome marker panel and 18 PCa-specific miRNAs selected from available literature and databases analysis.

### 4.5. Statistical Analysis

Statistical analysis was performed using the R environment [51]. Threshold cycle (*C*t) values were normalized using the pair ratio method. Significant combinations of miRNA pairs were selected by the RandomForest-based feature selection algorithm Boruta. A total of 3403 combinations were made. A number of miRNAs were missing in some samples. The following method was used to not exclude these miRNAs from the analysis. First, 9 ratio values of each microRNA pair were randomly selected from each group without replacement. Then, the significance of these combinations as predictors of PCa, HD and BPH was evaluated using Boruta. This procedure was repeated 1000 times, and significant predictors were recorded at each iteration. For the selected miRNA pairs, the difference of the medians between the groups was assessed and a 95% confidence interval was built using the bootstrap approach [52].

The miRNA combinations with the confidence interval containing zero were excluded from the further analysis. The asymptotic Wilcoxon-Mann-Whitney Test was performed for the remaining miRNA combinations to compare different groups of donors. A *p*-value  <  0.05 was considered statistically significant. Benjamini-Hochberg correction (*p*_adj_) was used to adjust the statistical significance for multiple comparisons. 

The specificity and sensitivity of the analytical systems were obtained using Receiving Operator Characteristic (ROC) curves. The Area under ROC curves (AUC) was used to assess the diagnostic performance of miRNA combinations. 

## 5. Conclusions

A random Forest-based feature selection (Boruta) was used to analyze miRNA expression in urine supernatant and EVs of healthy men and PCa and BPH patients. Several miRNAs were selected as candidate biomarkers based on criteria such as frequency, median distance and the *p*-value. The best combinations were chosen and their diagnostic potential was determined. This work is a preliminary stage. The next step is to validate the selected miRNA markers using an independent group of donors and investigate the association of miRNA expression with demographical and clinico-pathological factors, including age, disease stage, Gleason index, tumor genotype, resistance to chemo- and radiotherapy, patient survival, etc.

## Figures and Tables

**Figure 1 ijms-19-04088-f001:**
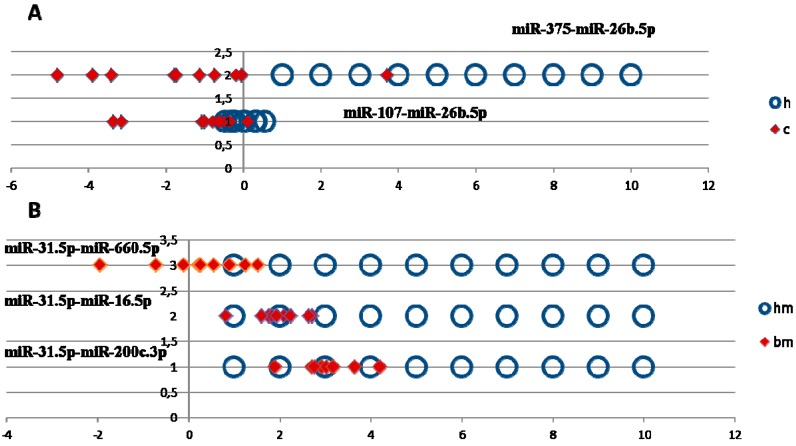
(**A**) Distribution of normalized d*C*_t_ values for miR-375.3p-miR-26b.5p and miR-107-miR-26b.5p pairs in healthy donors and PCa patients. (**B**) Distribution of normalized values of healthy donors and patients with BPH, miR-31.5p-miR-660.5p, miR-31.5p-miR-16.5p and miR-31.5p-miR-200c.

**Table 1 ijms-19-04088-t001:** Significant predictors of group comparison.

No.	miRNA Pairs	Frequency, %	Median Distance (95% CI)	AUC	*p*	*p* _adj_
h–c comparison, 1770 pairs						
1	miR-107-miR-26b.5p	74.8	−0.74 [−1.94;−0.33]	0.93	0.0012	0.0237
2	miR-93.5p-miR-29b.3p	36.3	−0.54 [−1.31;−0.26]	0.92	0.0019	0.0212
3	miR-22.3p-miR-30e.5p	36.0	−0.76 [−1.95;−0.23]	0.86	0.0065	0.0256
4	miR-375-miR-26b.5p	6.2	−1.71 [−3.92;−0.17]	0.83	0.0143	0.0249
5	miR-29b.3p-miR-205.5p	4.6	1.92 [0.25;2.81]	0.88	0.0041	0.0202
6	miR-331.3p-miR-205.5p	0.3	1.72 [0.24;3.06]	0.79	0.0018	0.0373
7	miR-205.5p-miR-26b.5p	15.2	−1.4 [−3.17;−0.55]	0.88	0.0041	0.0237
8	miR-29a.3p-miR-205.5p	10.3	1.69 [0.57;2.74]	0.83	0.0126	0.0252
9	miR-151a.5p-miR-205.5p	6.3	1.18 [0.43;3.29]	0.84	0.0102	0.0252
10	let-7e.5p-miR-23b.3p	15.8	0.88 [0.27;1.35]	0.92	0.0019	0.0237
h–b comparison, 254 pairs						
1	miR-30a.5p-let-7g.5p	98.5	0.57 [0.28;1.09]	0.83	0.0143	0.0429
2	miR-200a.3p-let-7a.5p	3.9	0.53 [0.03;1.34]	0.7	0.0054	0.1416
3	miR-205.5p-let-7a.5p	0.2	0.86 [0.06;2.71]	0.78	0.0412	0.0618
b–c comparison, 254 pairs						
1	miR-103a.3p-miR-30c.5p	28.6	1.03 [0.03;1.45]	0.79	0.0338	0.0495
2	hsa-miR-30c.5p-miR-30e.5p	21.8	−0.43 [−1.13;−0.17]	0.83	0.0143	0.0429
3	miR-100.5p-miR-30e.3p	18.9	0.73 [0.02;1.02]	0.77	0.05	0.05
4	miR-30c.5p-miR-16.5p	15.2	−0.74 [−1.36;−0.01]	0.84	0.0114	0.0429
5	miR-23b.3p-miR-103a.3p	8.8	−0.41 [−0.99;−0.03]	0.78	0.0412	0.0495
6	miR-100.5p-miR-30a.5p	7.2	0.52 [0.09;1.39]	0.79	0.0338	0.0495
hm–cm comparison, 849 pairs						
1	miR-20a.5p-miR-16.5p	50.3	0.39 [0.17;0.8]	0.89	0.0032	0.0245
2	miR-24.3p-miR-200b.3p	12.0	0.47 [0.13;1.08]	0.88	0.0055	0.0245
3	miR-101.3p-miR-30e.5p	11.8	−0.28 [−0.46;−0.09]	0.82	0.0179	0.0268
4	miR-24.3p-miR-16.5p	5.3	0.45 [0.09;0.95]	0.85	0.0082	0.0245
5	miR-99b.5p-miR-24.3p	3.5	−0.25 [−0.86;−0.06]	0.76	0.0102	0.0494
6	miR-30b.5p-miR-125b.5p	3.3	0.32 [0.06;0.91]	0.8	0.0233	0.03
7	miR-31.5p-miR-16.5p	1.7	0.77 [0.08;1.23]	0.85	0.0118	0.0267
8	miR-30c.5p-miR-16.5p	0.3	0.42 [0;0.9]	0.82	0.0156	0.0268
9	miR-30b.5p-miR-16.5p	0.1	0.85 [0.08;1.5]	0.79	0.0284	0.0319
hm–bm comparison, 849 pairs						
1	miR-107-miR-31.5p	95.3	−0.73 [−1.95;−0.27]	0.87	0.0071	0.0141
2	miR-31.5p-miR-16.5p	61.5	0.79 [0.31;1.25]	0.89	0.0043	0.0141
3	miR-31.5p-miR-30e.3p	43.8	0.69 [0.15;1.5]	0.88	0.0055	0.0141
4	miR-31.5p-miR-200b.3p	23.9	0.58 [0.18;1.3]	0.88	0.0055	0.0141
5	miR-31.5p-miR-660-5p	22.1	0.74 [0.08;1.79]	0.84	0.0023	0.0142
6	miR-29a.3p-miR-660.5p	20.4	0.65 [0.38;1.06]	0.86	0.009	0.0142
7	miR-31.5p-miR-200c.3p	17.3	0.63 [0.25;1.24]	0.83	0.0143	0.0159
8	miR-107-miR-141.3p	5.6	−0.79 [−1.95;−0.09]	0.78	0.0412	0.0412
9	miR-29a.3p-miR-30e.3p	1.6	0.82 [0.04;1.16]	0.84	0.0114	0.0142
10	miR-660.5p-miR-24.3p	0.8	−0.58 [−1.16;−0.05]	0.87	0.0071	0.0141
bm–cm comparison, 254 pairs						
1	miR-191.5p-miR-31.5p	22.7	0.41 [0.16;1.09]	0.91	0.0025	0.0143
2	miR-100.5p-miR-30d.5p	11.1	0.32 [0.15;1.27]	0.84	0.0114	0.0177
3	miR-106b.5p-miR-191.5p	7.4	−0.71 [−1.34;−0.29]	0.88	0.0041	0.0143
4	miR-93.5p-miR-22.3p	2.9	−0.4 [−0.75;−0.02]	0.78	0.0034	0.0412
5	miR-191.5p-miR-200b.3p	0.3	0.54 [0.02;1.21]	0.83	0.0126	0.0177
6	miR-22.3p-miR-92a.3p	0.1	0.44 [0.12;0.89]	0.78	0.0412	0.0412
7	miR-191.5p-miR-200a.3p	0.1	0.69 [0.17;1.13]	0.83	0.0126	0.0177

**Table 2 ijms-19-04088-t002:** The occurrence of individual miRNAs in pairs selected from the comparisons for urine supernatant and urine EVs.

miRNAs	h–c	h–b	b–c	hm–cm	hm–bm	bm–cm
miR-16.5p	?		●	●	●	
miR-22.3p	●					●
miR-24.3p				●	●	
miR-23b.3p	●		●			
miR-29a.3p	●				●	
miR-30a.5p		●	●			
miR-30c.5p			●	●		
miR-30e.5p	●		●	●	●	
miR-31.5p				●	●	●
miR-93.5	●					●
miR-100.5p			●			●
miR-107	●				●	
miR-200a.3p		●				●
miR-200b.3p				●	●	●
miR-205.5p	●	●				

**Table 3 ijms-19-04088-t003:** The occurrence of miRNAs in group comparisons.

Group Comparison	miRNA	The Occurrence of miRNA within the Comparison Group, %	The Total Number of miRNA Pairs in the Comparison Group
h–c	miR-205.5p	50	10
	miR-26b.5p	40	
	miR-29b.3p	20	
h–b	miR-30c.5p	100	3
b–c	miR-30c.5p	50	6
	miR-30e.5p	33.34	
	miR-100.5p	33.34	
	miR-103a.3p	33.34	
hm–cm	miR-16.5p	55.56	9
	miR-24.3p	33.34	
	miR-30b.5p	22.23	
hm–bm	miR-31.5p	60	10
	miR-660.5p	30	
	miR-107	20	
	miR-30e.3p	20	
	miR-29a.3p	20	
bm–cm	miR-191.5p	57.15	7
	miR-22.3p	28.58	

**Table 4 ijms-19-04088-t004:** Diagnostic performance of miRNA pairs in urine supernatant and urine EVs.

Group Comparison	miRNA Pairs	AUC	Threshold	95% CI	Sensitivity%	Specificity%
h–c	miR-205.5p-miR-26b.5p	0.88	<−3.365	0.7212–1.039	30	100
	miR-107-miR-26b.5p	0.93	<−0.4896	0.8074–1.053	80	100
	miR-375-miR-26b.5p	0.83	<−0.6851	0.6293–1.037	70	100
	miR-151a.5p-miR-205.5p	0.84	>3.152	0.6506–1.029	30	100
	miR-29b.3p-miR-205.5p	0.88	>2.034	0.7278–1.032	50	100
	miR-29a.3p-miR-205.5p	0.83	> 3.344	0.6368–1.023	20	100
	miR-331.3p-miR-205.5p	0.79	>5.651	0.5624–1.015	10	100
h–b	miR-30a.5p-let-7g.5p	0.83	<−2.323	0.6107–1.056	33	100
b–c	miR-30c.5p-miR-30e.5p	0.83	>−0.7914	0.6351–1.032	20	100
	miR-103a.3p-miR-30c.5p	0.79	<1.153	0.5727–1.005	60	100
hm–cm	miR-20a.5p-miR-16.5p	0.85	<0.7185	0.6502–1.047	56	100
	miR-30b.5p-miR-16.5p	0.83	<−2.959	0.6355–1.021	22	100
	miR-31.5p-miR-16.5p	0.79	<1.833	0.5630–1.012	37.5	100
	miR-24.3p-miR-200b.3p	0.93	<1.307	0.8231–1.044	67	100
hm–bm	miR-31.5p-miR-200c.3p	0.83	<3.063	0.6356–1.031	60	100
	miR-31.5p-miR-16.5p	0.89	<2.256	0.7312–1.047	80	100
	miR-107-miR-141.3p	0.78	>5.213	0.5454–1.010	10	100
	miR-31.5p-miR-200b.3p	0.88	<3.104	0.7168–1.039	70	100
	miR-31.5p-miR-30e.3p	0.88	<1.252	0.7067–1.049	80	100
	miR-29a.3p-miR-30e.3p	0.84	<−0.1366	0.6625–1.026	40	100
	miR-31.5p-miR-660.5p	0.84	<0.5698	0.6630–1.026	70	100
	miR-29a.3p-miR-660.5p	0.86	<−0.7334	0.6679–1.043	20	100
	miR-20a.5p-miR-16.5p	0.73	<0.8182	0.4669–0.9931	70	100
	miR-107-miR-31.5p	0.87	>1.324	0.6716–1.062	10	100
bm–cm	miR-191.5p-miR-200a.3p	0.83	<2.595	0.6479–1.012	40	100
	miR-191.5p-miR-31.5p	0.91	<0.07563	0.7707–1.052	33	100
	let-7i.5p-let-7a.5p	0.83	<4.612	0.6117–1.039	57.14	100
	miR-100.5p-miR-200b.3p	0.81	<3.867	0.6002–1.022	55.56	100
	miR-106b.5p-miR-191.5p	0.88	>0.3940	0.7234–1.037	30	100

**Table 5 ijms-19-04088-t005:** Participation of identified predictors in the development and progression of prostate cancer according to the OncomiRDB database.

miRNA	miRNA Function	Sum Effect	Direct Targets
miR-16.5p	Reduce cell proliferation	Tumor-suppressive	FGF2
Inhibit tumor growth	FGFR1
miR-20a.5p	Increase cell proliferation	Oncogenic	CX43
Increase colony formation	
miR-23b.3p	Inhibit cell proliferation	Tumor-suppressive	
Inhibit colony formation	
Inhibit cell migration	
Inhibit cell invasion	SRC
Induce cell cycle G0/G1 arrest	AKT1
Induce apoptosis	
Inhibit epithelial-mesenchymal transition	
miR-26b.5p	Inhibit cell proliferation	Tumor-suppressive	
miR-29a.3p	Inhibit cell growth		TRIM68
Inhibit cell invasion	
miR-29b.3p	Inhibit wound healing	Tumor-suppressive	
Inhibit cell invasion
Inhibit colony formation
miR-30d.5p	Promote cell proliferation	Oncogenic	SOCS1
Promote cell invasion	
miR-101.3p	Inhibit cell growth	Tumor-suppressive	COX2
Inhibit tumor growth	
miR-106b.5p	Increase cell adhesion	Oncogenic	CASP7
Promote cell growth	
miR-125b.5p	Promote tumor growth, reduce drug sensitivity	Oncogenic	TP53 BBC3BAK1
miR-141.3p	Enhance cell growth	Oncogenic	
miR-200a.3p	Inhibit cell invasion	Tumor-suppressive	GNA13
miR-200b.3p	Inhibit cell migration	Tumor-suppressive	
Inhibit cell invasion
Reduce cell adhesion
Reduce cell detachment
miR-200c.3p	Increase adhesion	Tumor-suppressive	
Reduce cell invasion
Reduce cell migration
miR-205.5p	Increase adhesion	Tumor-suppressive	ZEB1
Reduce cell invasion
Reduce cell migration
Inhibit epithelial-mesenchymal transition	VIM
Inhibit cell migration

miR-31.5p	Inhibit cell proliferation	Tumor-suppressive	
Inhibit cell invasion
Inhibit cell migration
miR-99b.5p		Tumor-suppressive	SMARCA5
Inhibit cell growth	SMARCD1
	MTOR
miR-100.5p		Tumor-suppressive	SMARCA5
Inhibit cell growth	SMARCD1
	MTOR
miR-331.3p	Inhibit cell proliferation	Tumor-suppressive	DOHH

**Table 6 ijms-19-04088-t006:** Clinicopathological and demographic characteristics of healthy donors and patients with BPH and cancer.

Donor Characteristics	HD	BPH	PCa
	*n* = 10	*n* = 10	*n* = 10
**Age**			
Average	57.5	68	70.5
Median [1;3 quartile]	57 [52;64]	66 [61;78]	74 [65;76]
Range	48–66	53–81	56–82
**Total PCA, ng/mL**			
Average	0.817	11.6	16.3
Median [1;3 quartile]	0.91 [0.4;1.1]	8.5 [5;10.9]	14 [11.2;21.3]
Range	0.13–1.54	1.97–41.5	9.24–26.4
**PCa stage**			T2-3NxMx
**T2**			5 (50%)
**T3**			5 (50%)
**Nx**			10 (100%)
**Mx**			10 (100%)
**Gleason score 6**			5 (50%)
**Gleason score 7**			5 (50%)

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
