# Peer review of "Searching for the Novel Specific Predictors of Prostate Cancer in Urine: The Analysis of 84 miRNA Expression"

_ijms, 2018, doi:10.3390/ijms19124088_

Round 1
Reviewer 1 Report
Dr. Lekchnov and collegues analyzes and compares in this manuscript miRNA profiles of clarified urine supernatant and combined urine vesicle fractions of healthy donors, patients with benign prostatic hyperplasia (BPH) and PC.
The authors made a very good statistical analysis of their data and present their data thanks to good tables and figures.
The paper is well written, only a small revision of the english language is required and some sentences are not in black but in grey.
The minor concern is that the authors should pay attention to write in full the first time in the text Ct values (line 107), EVs, BPH.
Moreover they should rewrite the sentence at lines 107-111 that is a little tortuous.
Major point:
The authors show a lot of tables in results section. The results are written in a very technical way by a statistical point of view. It's appreciable, but from the whole paragraph it is very difficul to understand clearly and easily what are the most salient results.
Moreover, also if the tables and the figures are very well performed, it is not clear if the miRNA belonging to healthy patients, BPH patient and PC patients are different or significantly differently expressed.
Moreover, the authors should specify clearly, through their results what are the differences observed between clarified urine supernatant and combined urine vescicle fractions.
The authors should exalt this points, especially for the future readers which could be also not directly research insiders.
Only an additional curiosity: why the authors did not consider patients affected by PC with a Gleason score >7?
Author Response
Reviewer 1:
Dear Reviewer,
We sincerely appreciate your valuable advices and comments. They were extremely useful and
helped us substantially improve our manuscript.
The minor concern is that the authors should pay attention to write in full the first time in the text Ct values (line 107), EVs, BPH.
The phrase “Ct values” has been corrected by “threshold cycle (Ct) values”. The acronyms EVs, BPH have also been properly introduced.
Moreover they should rewrite the sentence at lines 107-111 that is a little tortuous.
This and several other inappropriately complicated sentences throughout the manuscript were adjusted.
The authors show a lot of tables in results section. The results are written in a very technical way by a statistical point of view. It's appreciable, but from the whole paragraph it is very difficult to understand clearly and easily what are the most salient results. Moreover, also if the tables and the figures are very well performed, it is not clear if the miRNA belonging to healthy patients, BPH patient and PC patients are different or significantly differently expressed. Moreover, the authors should specify clearly, through their results what are the differences observed between clarified urine supernatant and combined urine vesicle fractions. The authors should exalt this points, especially for the future readers which could be also not directly research insiders.
The results were supplemented by a new segment starting at line 196, underlining the main results of the study and highlighting the statistically significant differences in the expression of miRNA between the three different groups of donors.
Only an additional curiosity: why the authors did not consider patients affected by PC with a Gleason score >7?
Our samples collection of did contain some samples from cancer patients with a Gleason score of more than 7 at the time of the study. However, since the number of such samples was too low to include them into the study alongside patients with lower Gleason scores, we decided not to include them in the present study. In future we are aiming to verify our results using a more representative sample including PCa patents with high Gleason scores.
Thank you very much for your point of view and revising of the manuscript. Your comments helped us greatly to improve our article.
Best regards,
on behalf of coauthors, Evgeniy A. Lekchnov.
Reviewer 2 Report
In this manuscript submitted by Lekchnov et al., authors analyze the expression of miRNA expression profiles of urine samples of healthy donors, patients with benign prostatic hyperplasia and prostate cancer. Still being a preliminary study, the data obtained with a comparative analysis reinforce the idea that specific miRNA could be potential biomarkers in prostate cancer. The experiments in this manuscript appear to be carefully done and they are nicely described.
Unfortunately, there are several shortcomings in the data presentation. The abstract is a mere description of the methods employed in this study. I believe that a brief description (1-2 lines) of the results obtained is needed. The introduction can be improved changing/removing some references and emphasizing the role of miRNAs as potential biomarkers. Finally, the results are too descriptive and a little hard to follow. In my opinion the authors should complete the results with a brief discussion/conclusion. On the other hand, I found the final discussion properly written.
Line 34: (https://www.cancer.org/content/cancer/en/cancer/prostate-cancer/detection-diagnosis staging/survival-rates.html). I have not been able to find the website because the following notice appears: “We're sorry. That page couldn’t be found. We are always updating information on cancer.org, so it’s possible that the page you’re looking for has been changed or moved”. It should be relatively simple for the authors replace this link with and appropriate reference such as a review.
Line 59: “Indeed, PSA doesn’t meet modern criteria for cancer biomarkers”. Please clarify what do you mean with “modern criteria”
Minor points:
Line 17: PCa should be “Prostate cancer (PCa)”, because it is the first time you introduce this terminology.
Line 26: “prostate cancer should be “PCa”
Line 40: maybe you want to say “magnetic resonance imaging (MRI)” ?
Line 44: What do you mean with “T2 and T3 tumours”?
Line 47: Please define “low risk PCa”. Most of the readers of this journal are not cancer experts
Line 51: is “AUC” the area under the curve ? Please define it
Line 54: “prostate-specific antigen” should be “PSA”
Line 74: “prostate cancer” should be “PCa”
Line 102: “In the pilot study” should be “In this/in the present…..”
Line 288: What does “TNM stage” mean?
Author Response
Reviewer 2:
Dear Reviewer,
we appreciate the critical appraisal of our manuscript and thank you for your comments.
The abstract is a mere description of the methods employed in this study. I believe that a brief description (1-2 lines) of the results obtained is needed.
We agree with your comment. We have modified the abstract to reflect the results of our study.
The introduction can be improved changing/removing some references and emphasizing the role of miRNAs as potential biomarkers.
We agree with you. The additional references have been provided to support the role of miRNA as cancer biomarkers.
Finally, the results are too descriptive and a little hard to follow. In my opinion the authors should complete the results with a brief discussion/conclusion. On the other hand, I found the final discussion properly written.
We agree with your comment. The results were supplemented with a conclusion, showing the main results of our study. The new section starts with line 196.
Line 34: (https://www.cancer.org/content/cancer/en/cancer/prostate-cancer/detection-diagnosis staging/survival-rates.html). I have not been able to find the website because the following notice appears: “We're sorry. That page couldn’t be found. We are always updating information on cancer.org, so it’s possible that the page you’re looking for has been changed or moved”. It should be relatively simple for the authors replace this link with and appropriate reference such as a review.
The link has been replaced and is currently available for viewing.
Line 59: “Indeed, PSA doesn’t meet modern criteria for cancer biomarkers”. Please clarify what do you mean with “modern criteria”
Our rationale here was that the specificity and sensitivity of PSA does not hold up with the rising standards of personalized medicine, which we feel is consistent with the wording of the manuscript. The sentence in question was changed to: “Due to this, PSA doesn’t meet modern criteria for an effective cancer biomarker and a replacement is overdue” to better reflect that notion.
Minor points:
Line 17: PCa should be “Prostate cancer (PCa)”, because it is the first time you introduce this terminology.
The phrase has been corrected.
Line 26: “prostate cancer should be “PCa”
The phrase has been corrected.
Line 40: maybe you want to say “magnetic resonance imaging (MRI)” ?
Right, thank you.
Line 44: What do you mean with “T2 and T3 tumours”?
We meant T2,3 stage tumors. The phrase “T2 and T3 tumours” has been corrected by “T2 and T3 stage tumours”
Line 47: Please define “low risk PCa”. Most of the readers of this journal are not cancer experts
The phrase “PSA < 10 ng/mL, and Gleason score < 7 and cT1-2a” has been added.
Line 51: is “AUC” the area under the curve ? Please define it
Right, “AUC” is the area under the curve. The sentence has been amended accordingly
Line 54: “prostate-specific antigen” should be “PSA”
Right, thank you.
Line 74: “prostate cancer” should be “PCa”
Right, thank you.
Line 102: “In the pilot study” should be “In this/in the present…..”
The phrase “In the pilot study” has been replaced by “In this study”
Line 288: What does “TNM stage” mean?
The phrase “PCa with 2-3 TNM stage” has been replaced by “PCa with T2-3NxMx stage and pathological Gleason score 6-7”
The manuscript was revised according to you very helpful comments.
Thank you also for the recommendations contributing to the manuscript language, the
manuscripts text was amended in accordance with them.
Best regards,
on behalf of coauthors, Evgeniy A. Lekchnov.
Round 2
Reviewer 2 Report
I'm satisfied with the authors corrections. Thank you